# Genetic Diversity of *Leuconostoc mesenteroides* Isolates from Traditional Montenegrin Brine Cheese

**DOI:** 10.3390/microorganisms9081612

**Published:** 2021-07-28

**Authors:** Werner Ruppitsch, Andjela Nisic, Patrick Hyden, Adriana Cabal, Jasmin Sucher, Anna Stöger, Franz Allerberger, Aleksandra Martinović

**Affiliations:** 1Institute of Medical Microbiology and Hygiene, Austrian Agency for Health and Food Safety, 1090 Vienna, Austria; adriana.cabal-rosel@ages.at (A.C.); jasmin.sucher@ages.at (J.S.); anna.stoeger@ages.at (A.S.); franz.allerberger@ages.at (F.A.); 2Centre of Excellence for Digitalization of Microbial Food Safety Risk Assessment and Quality Parameters for Accurate Food Authenticity Certification (FoodHub), University of Donja Gorica, 81000 Podgorica, Montenegro; andjela.nisic@udg.edu.me; 3Department of Microbiology and Ecosystem Science, University of Vienna, 1090 Vienna, Austria; patrick-christian.hyden@ages.at

**Keywords:** *Leuconostoc mesenteroides*, diversity, Montenegro, traditional cheese, whole genome sequencing, virulence, antimicrobial resistance, flavor, food safety

## Abstract

In many dairy products, *Leuconostoc* spp. is a natural part of non-starter lactic acid bacteria (NSLAB) accounting for flavor development. However, data on the genomic diversity of *Leuconostoc* spp. isolates obtained from cheese are still scarce. The focus of this study was the genomic characterization of *Leuconostoc* spp. obtained from different traditional Montenegrin brine cheeses with the aim to explore their diversity and provide genetic information as a basis for the selection of strains for future cheese production. In 2019, sixteen *Leuconostoc* spp. isolates were obtained from white brine cheeses from nine different producers located in three municipalities in the northern region of Montenegro. All isolates were identified as *Ln.* *mesenteroides.* Classical multilocus sequence tying (MLST) and core genome (cg) MLST revealed a high diversity of the Montenegrin *Ln. mesenteroides* cheese isolates. All isolates carried genes of the bacteriocin biosynthetic gene clusters, eight out of 16 strains carried the *cit*CDEFG operon, 14 carried *butA,* and all 16 isolates carried *alsS* and *ilv*, genes involved in forming important aromas and flavor compounds. Safety evaluation indicated that isolates carried no pathogenic factors and no virulence factors. In conclusion, *Ln. mesenteroides* isolates from Montenegrin traditional cheeses displayed a high genetic diversity and were unrelated to strains deposited in GenBank.

## 1. Introduction

The genus *Leuconostoc* (*Ln.*) currently comprises 14 species and eight subspecies, all being Gram-positive, non-spore-forming, non-motile Lactic Acid Bacteria (LAB). Within the genus *Leuconostoc*, *Ln. mesenteroides*, *Ln. pseudomesenteroides* and *Ln. lactis* have their role in food fermentation and can be isolated from various food-related ecological niches, including beverages, meat and dairy products, and some plant materials, implying wide distribution and specialized adaptation to these diverse environments [1,2,3,4]. The first description of *Ln. mesenteroides* was by Van Tieghem in 1878 and was later proposed as the type strain [5]. In 1983, *Ln. dextranicum* and *Ln. cremoris* were reclassified as subspecies of *Ln. mesenteroides* due to the common properties they shared [6] and their high degree of relatedness shown by DNA—DNA hybridization [5].

In the food production, *Leuconostoc* spp. are usually applied as an adjunct culture in combination with the fast acid producing *Lactococcus* spp., as undefined mixed (DL) type starter cultures, contributing to aroma and texture formation of the final products through citrate degradation and production of diacetyl, acetoin, and carbon dioxide [7,8]. Due to its aroma producing traits, *Ln. mesenteroides* has become known as “the aroma bacterium” [9]. *Leuconostoc* spp. are characterized by heterolactic fermentation [10] through the phosphoketolase pathway (PKP), resulting in the formation of aroma and flavor compounds such as lactic acid, acetic acid, ethanol, and CO_2_ (responsible for “eye formation” in some cheeses) [8]. Genetic elements involved in the production of aroma and flavor compounds include the acetoin reductase gene *butA*, the citrate inducible *cit* operon, and genes involved in the biosynthesis of branched aminoacids like the acetolactate synthase gene *alsS* and genes of the isoleucine-leucin (*ilv*) synthesis operon. These compounds contribute to the flavor of the products, while biosynthesis of bacteriocins [11,12,13] has preservation potential by inhibiting the growth of pathogenic bacteria like *Listeria* spp., *Escherichia coli*, *Staphylococcus aureus*, or *Salmonella enterica* serovars Typhimurium and Typhi [11,12,13], positively affecting the quality and safety of the final product [14]. 

Many traditional dairy products are manufactured without starter cultures [15] and are recognized as an essential part of a country’s identity, culture, and heritage [16,17]. In these traditional dairy products, naturally occurring *Leuconostoc* spp., together with mesophilic lactobacilli known as non-starter LAB (NSLAB), play an important role in flavor development, accounting for the characteristics and the quality of traditional food products [15]. In addition, *Leuconostoc* spp. strains obtained from traditional cheese products can be valuable for future dairy food production [17], by improving the product quality [18], since their properties ultimately affect the characteristics and quality of the final products. Therefore, the isolation and accurate characterization of *Leuconostoc* spp. strains, as well as other NSLAB species, can provide necessary information for their selection and application with the aim of preservation and improvement of quality and safety of traditional food products [17]. With the evolution of sequencing technologies from Sanger sequencing to whole genome sequencing (WGS), the genome-based characterization of *Leuconostoc* spp. may present an innovative tool to ensure consistent manufacturing of high quality and safe traditional food products [19,20]. Despite the proven influence of *Leuconostoc* spp. to the final characteristics of cheeses, information on the genomic diversity of *Leuconostoc* spp. obtained from cheese is still scarce [20,21,22]. Up to now, three different multilocus sequence typing [MLST] schemes have been published to investigate strain diversity of approximately 200 isolates from environmental and dairy sources [23,24,25]. Among all those 200 isolates, only seven cheese isolates were analyzed by MLST [23,24,25]. Unfortunately, publicly available databases do not exist for any of the three published MLST schemes [23,24,25], making a direct strain comparison impossible. In addition, in today’s genomic era, only seven *Ln. mesenteroides* genomes (DSM20484, FM06, LbT16, LbE15, LbE16, LN25 and T26) derived from cheese are available in GenBank (accessed 15 June 2021) [20,21,22,26,27]. However, genome sequencing remains the most powerful tool for characterization of bacterial strains. Prospective comparison and coupling of genome data and other “omics” data [19,28] with phenotypic data and the use of genome-scale metabolic models [GSMMs] may facilitate the prediction of metabolic patterns and easier comparison of metabolic capacities of organisms [8]. The combination of all these modern technologies may assist the rapid and efficient genetic marker-based selection of the most suitable strains even though predictive microbiology is currently still a complex and challenging process [8,19,29].

The aim of this study was to explore the genetic diversity of *Ln. mesenteroides* isolates obtained from Montenegrin brine cheeses using whole genome sequencing to prove the origin of the Montenegrin strains that can be used in future industrial cheese production. This could lead to more controlled cheese production featuring improved product safety with the preserved indigenous sensory characteristics as an added value. The use of these genome data could represent a useful tool for authentication of traditional dairy products.

## 2. Materials and Methods

### 2.1. Origin and Cultivation of Isolates

Bacteria were isolated from 13 different white brine ripened traditional cheeses collected during spring and summer 2019 from nine producers from the three municipalities Pljevlja, Šavnik, and Žabljak in the northern region of Montenegro (Appendix A Regions of Montengro), as described previously [30]. Briefly, for each sample, 20 g of cheese was added to 180 mL sterile 2% (*w*/*v*) sodiumcitrate solution and homogenized for 2 min in an Omni mixer (Omni International, Waterbury, CT, USA). For enumeration, serial dilutions in Ringer’s solution were plated on de Man, Rogosa, Sharpe (MRS), and M17 agar plates (HiMedia, Mumbai, India) and incubated at 30 °C for 72 h. Isolates were biochemically characterized for their acidification and post-acidification ability, growth at different temperatures and NaCl concentrations, and their ability for lactose degradation. Isolates were stored in MRS and M17 medium supplemented with 15% glycerol (*v*/*v*) at −80 °C, and, for revitalization, subcultured three times in the respective broths (MRS and M17) at 30 °C overnight.

Up to ten single colonies were selected from MRS and/or M17 agar plates and subcultured for further processing. Species confirmation was carried out using matrix assisted laser desorption/ionization time-of-flight mass spectrometry (MALDI-TOF-MS) [31] on a MALDI-TOF Microflex LT/SH with database MBT Compass IVD 4.2 according to the manufacturer’s instructions (Bruker, Billerica, MA, USA).

### 2.2. Biochemical Testing

#### 2.2.1. Growth at Different Temperatures

Overnight cultures grown at 30 °C were inoculated in MRS broth and incubated for 24 h at 4, 10, 15, 30 (optimal growth temperature, used as control), and 45 °C. The optical density was measured on a Jasco UV/VIS spectrophotometer V-730 (Jasco, Cremella, Italy) at 560 nm (OD_560_).

#### 2.2.2. Production of CO_2_

The production of CO_2_ from glucose was determined as described previously [32]. Briefly, bacterial cultures (50 µL) were added into a test tube containing 10 mL of suitable broth supplemented with 1% of glucose (Tokyo Chemical Industry, Tokyo, Japan) and a Durham’s tube. After 24–48 h of incubation at 30 °C, gas production was assessed: if the gas accumulated in the Durham’s tube to more than one third of its capacity, the result was considered positive.

#### 2.2.3. Salt Tolerance

The evaluation of salt tolerance was performed as described previously [32,33]. Briefly, salt content of tubes containing 3 mL of MRS broth was adjusted to a final concentration of 2%, 3%, 4.5%, or 6.5% (*w*/*v*) NaCl. Tubes were inoculated with overnight cultures of the strains and incubated at 30 °C for 24 h. Bacteria grown in MRS broth without NaCl were used as controls. The optical density was measured on a Jasco UV/VIS spectrophotometer V-730 at 560 nm (OD_560_).

#### 2.2.4. Acidification and Post-Acidification Ability in Milk

Acidification and post acidification ability was tested as described previously by inoculating 50 mL ultra-high temperature (UHT) processed skimmed cow’s milk (Imlek, Belgrade, Serbia) [34] with 50 µL of broth culture and incubated at 30 °C for 48 h. The resulting pH values, measured after 2, 4, 6, 8, 12, and 24 h post-acidification, were investigated by measuring the pH of the inoculated skimmed milk after 48 h of incubation [34]. pH of milk was measured with a WTW pH Meter inoLab ph 7110 (Xylem Analytics, Weilheim, Germany).

#### 2.2.5. Catalase Testing

Bacterial cells from a solid nutrient medium were transferred to empty Petri dishes. In addition, 1–2 drops of 3% hydrogen peroxide were dropped on the cells. Formation of air bubbles (production of oxygen due to the presence of catalase) was examined visually.

#### 2.2.6. Ability for Formation of Exopolysaccharides (EPS)

EPS formation ability was performed by identifying mucous-producing colonies grown on MRS agar plates. Additionally, mucoid colonies were stretched with an inoculation loop to detect the production of long filaments. If the mucus producing colonies or the formation of long filaments was missing, the strain was considered EPS negative.

### 2.3. Antimicrobial Susceptibility Testing

The Kirby–Bauer disk diffusion test was used to determine antimicrobial susceptibility of strain INF36 against erythromycin (15 µg, Bioanalyse, Ankara, Turkey) and of strain INF117 against ampicillin (10 µg) and penicillin (1U) (Bioanalyse, Ankara, Turkey). The disk diffusion patterns were evaluated according to the microbiological breakpoints for selected lactic acid bacteria as defined by EFSA [35,36].

### 2.4. DNA Extraction and Whole Genome Sequencing

For whole genome sequencing, high quality genomic DNA was isolated from overnight cultures grown in M17 or MRS using the MagAttract HMW DNA Kit (Qiagen, Hilden, Germany) according to the protocol for Gram-positive bacteria following the manufacturer’s instructions (Qiagen). The amount of input DNA was quantified on a Lunatic instrument (Unchained Labs, Pleasanton, CA, USA). Ready to sequence libraries were prepared using the Nextera XT DNA library preparation kit (Illumina, San Diego, CA, USA) according to the instructions of the manufacturer. Paired-end sequencing with a read length of 2 × 300 bp using Reagent Kit v3 chemistry (Illumina, San Diego, CA, USA) was performed on a Miseq instrument (Illumina, San Diego, CA, USA) according to the instructions of the manufacturer (Illumina).

### 2.5. Sequence Data Analysis

All study isolates were sequenced to obtain a coverage of at least 20-fold. Raw reads were quality controlled using FastQC v0.11.9 and de novo assembled using SPAdes (version 3.11.1) [37] to produce draft genomes. Contigs were filtered for a minimum coverage of 5× and a minimum length of 200 bp using SeqSphere+ software v7.2.3 (Ridom, Münster, Germany). Confirmation of species identification was done by 16S rRNA analysis [38], https://tygs.dsmz.de (accessed on 27 November 2020), Mash distance v2.3 [39], and ribosomal multilocus sequence typing (rMLST) [40]. Average nucleotide identity (ANI) was determined using FastANI v1.32 [41] in all-versus-all fashion using default parameters. A Type Strain Genome Server analysis (TYGS) tool from the German Collection of Microorganisms and Cell Cultures GmbH (DSMZ) was used for digital DNA-DNA hybridization (dDDH) using formula d4 as recommended for draft genomes (https://tygs.dsmz.de (accessed on 27 November 2020)) to assign the isolates to *Ln. mesenteroides* subspecies [40], https://tygs.dsmz.de (accessed on 27 November 2020).

Subtyping of all 16 *Ln. mesenteroides* isolates was conducted in SeqSphere+ v7.2.3 using a recently published eight-gene multilocus sequence typing (MLST) scheme comprising the eight loci *carB*, *groeL*, *murC*, *pheS*, *pyrG*, *recA*, *rpoB*, and *uvrC* [25] and a newly defined ad hoc core genome multilocus sequence typing (cgMLST) scheme. For definition of the cgMLST scheme, a genome-wide gene-by-gene comparison was performed using the complete genome of strain *Ln. mesenteroides* subsp. *mesenteroides* ATCC 8293 as a reference genome and 18 complete genomes of *Ln. mesenteroides* available in GenBank as query genomes (Appendix A). The resulting ad hoc cgMLST scheme comprised 960 core genome target genes and 935 accessory genome target genes. Sixty genes of the ATCC 8293 genome were discarded since they did not fulfil the default quality criteria (Appendix A). Seven of the eight MLST targets [25] were also part of our cgMLST scheme while one target (*uvrC*) belonged to the accessory genome. For phylogenetic analysis, a minimum spanning tree (MST) was calculated based on the defined 960-target cgMLST scheme. Based on the 17 allelic differences observed between the query strains NBRC107766 and ATCC19254 (both are the same strain but originating from different repositories and sequenced at different laboratories), an arbitrary complex type threshold of 45 allelic differences was applied for detection of related isolates.

BAGEL4 [42] was used to screen for bacteriocins, and antiSMASH5 [43] was used to screen for secondary metabolite “biosynthetic gene clusters”. To address biosafety concerns towards starter culture microorganisms, all 16 isolates were analyzed for the presence of pathogenic factors [44], plasmids [45], and virulence genes [46] using PathogenFinder 1.1, PlasmidFinder 2.1, and VirulenceFinder 2.0 available from the Center for Genomic Epidemiology [http://www.genomicepidemiology.org/ (accessed on 3 December 2020)]. Antibiotic resistance genes were detected using the ResFinder 4.1 tool with default settings (>90% identity and >60% coverage) [47] and the Comprehensive Antibiotic Resistance Database (CARD) applying perfect and strict hits search criteria [48].

### 2.6. Gene Presence and Absence

Detailed analysis of additional genomic information was performed for all 16 genomes and 12 reference strains: KMB608, ATCC8293, 406, 213M0, WC0331, DRC1506, ATCC19254, LN08, DSM20484, LN32, NBRC100495, and LbE15. The contigs of each assembly were filtered for a minimum length of 1000 nucleotides. Genes were predicted using prodigal v2.6.3 [49] with default parameters, and orthologous groups were calculated with OrthoFinder v1.4.2 [50]. Orthologous groups with differences in presence/absence were selected for visualization. Additionally, a species tree was created from the orthologous groups using default parameters for Orthofinder. Annotation of predicted genes was performed using NCBI BLAST+ v2.10.0 [51].

### 2.7. Nucleotide Sequence Accession Numbers

This *Leuconostoc mesenteroides* whole genome shotgun (WGS) project has been deposited in DDBJ/ENA/GenBank under the BioProject No. PRJNA706746. This Whole Genome Shotgun project has been deposited at DDBJ/ENA/GenBank under the accession JAGFYI000000000—JAGFYL000000000, JAFREA000000000—JAFREC000000000, and JAFRDS000000000—JAFRDZ000000000. The version described in this paper is version JAGFYI010000000—JAGFYL010000000, JAFREA010000000—JAFREC010000000, and JAFRDS010000000—JAFRDZ010000000.

## 3. Results

### 3.1. Biochemical Properties

Sixteen *Ln. mesenteroides* isolates were successfully isolated and identified by MALDI-TOF-MS from 13 Montenegrin brine cheeses (Appendix A Regions of Montenegro, Table 1). All isolates were Gram-positive, catalase negative, and had no ability for formation of exopolysaccharides (EPS). The mean number of viable bacterial cells was in the range between 41 to 92 cfu/g cheese. The acidification ability of the tested strains was in the range of pH 4.3–5.9 (24 h), while post acidification ability was between pH 4.2–5.7 (48 h). After 48 h of incubation at 30 °C, the pH in broths supplemented with 1% lactose was in the range of 4.4 to 5.5. Six tested isolates exhibited an ability to grow at 6.5% NaCl concentration, 11 strains at 4.5%, while three strains did not grow at 2% NaCl concentration. Both isolates obtained from producer A (INF2 and INF3b) showed growth ability at a salt concentration of 2%, but did not grow at higher salt concentrations. Isolate INF82b (producer D) was only growing at a salt concentration of 4.5%. Ten isolates showed an ability to grow at 45 °C, three isolates INF3b (producer A), INF36 (producer B), and INF46 (producer C) had the ability to grow at 4 °C (Table 1).

### 3.2. Species and Subspecies Identification

Mash distance analysis, 16s rRNA gene sequence analysis (Appendix A 16S rRNA), and ribosomal multilocus sequence typing (rMLST) (https://pubmlst.org/rmlst (accessed on 20 November 2020)) and ANI analysis (>97% identity; cut-off >95% identity) (Appendix A) confirmed the MALDI-TOF-MS identification of all 16 isolates as *Ln. mesenteroides*. Type Strain Genome Server analysis (TYGS) assigned 10 isolates to the subspecies *Ln. mesenteroides mesenteroides* and two each of the isolates to subspecies *Ln. mesenteroides dextranicum*, *Ln. mesenteroides cremoris*, and *Ln. mesenteroides jonggajibkimichii* (Figure 1). Digital DNA-DNA Hybridization (dDDH) using formula d4 (https://tygs.dsmz.de (accessed on 20 November 2020)) revealed a similarity of ≥92.5% of the ten *Ln. mesenteroides mesenteroides* to *Ln. mesenteroides mesenteroides* ATCC 8293 (Appendix A). Two isolates showed a dDDH similarity to subspecies *Ln. mesenteroides dextranicum* DSM 20484 of ≥97.8%, while four isolates were not clearly assignable to a subspecies by dDDH: INF2 and INF98 had dDDH values of 90.5% to *Ln. mesenteroides cremoris* and ≥93% to *Ln. mesenteroides dextranicum,* and INF117 and INF166 had dDDH values of ≥91.5% to *Ln. mesenteroides jonggajibkimichii*, ≥92.4% to *Ln. mesenteroides dextranicum*, and ≥92.6% to *Ln. mesenteroides mesenteroides* (Appendix A). ANI analysis (>97% identity, cut-off >95% identity) (Appendix A, Appendix A) and core genome analysis confirmed the dDDH results; cgMLST positioned these isolates on branches comprising strains of different subspecies (Figure 2). The Montenegrin cheese *Ln. mesenteroides* isolates had a genome size between 2.0 Mb and 2.3 Mb, had 2074 to 2368 genes, 1905 to 2221 coding genes, 85 to 151 pseudo genes, and 57 to 65 RNA genes (Appendix A).

### 3.3. Whole Genome Sequence Based Subtyping

MLST and cgMLST based characterization of all 16 *Ln. mesenteroides* isolates and 72 *Ln. mesenteroides* genomes available in GenBank (72 genomes fulfilled the quality criteria of at least ≥90% good core genome targets) revealed a high diversity of the Montenegrin *Ln. mesenteroides* isolates (Figure 2, Appendix A). MLST and cgMLST showed comparable results but with the advantage of a higher resolution for the core genome approach.

Briefly, the 16 Montenegrin brine cheese isolates were assigned to twelve different MLST profiles differing between zero and eight alleles. Identical MLST profiles were found for INF2 and INF98 from producers A and D (MLST complex 1); INF82b and INF94 both from producer D (MLST complex 2); INF46 and INF157 from producers C and H (MLST complex 3); INF36, INF90 and 213MO from producers B and D and GenBank, respectively (MLST complex 4); and INF103 from producer E and M11, 406, and AtHG50 from GenBank (MLST complex 5) (Appendix A). CgMLST differentiated the Montenegrin isolates by one to 860 alleles from each other (Figure 2). CgMLST confirmed the MLST based clustering of complex 1 to complex 3 when applying a core genome cluster type threshold of 45 alleles (Figure 2, Appendix A). MLST complex 1 isolates INF2 and INF98, MLST complex 2 isolates INF82b and INF94, and MLST complex 3 isolates INF46 and INF157 differed by 41, five, and one allele(s) in their core genome, respectively (Figure 2). CgMLST revealed with an allelic distance of 41 a closer relatedness of isolate INF90 to the MLST complex 2 isolates INF94 and INF82b than to the MLST complex 4 isolates INF36 and 213MO with an allelic distance of >170 alleles (Figure 2). For MLST complex 5 cgMLST analysis differentiated isolate INF103 from 138 to 163 alleles from the GenBank isolates 406, M11, and AtHG50. Isolates INF2 and INF3 obtained from two cheeses of the same type from producer A differed by 839 alleles in their core genome (Figure 2). Six isolates INF82a, INF82b, INF90, INF92, INF94, and INF98 obtained from three cheeses of the same type from producer D differed by five to a maximum of 860 alleles in their core genome (Table 1, Figure 2). Isolates INF82a and INF82b obtained from the same cheese from producer D differed by 860 alleles (Figure 2). The cgMLST based comparison of the Montenegrin *Ln. mesenteroides* isolates with isolates available from GenBank revealed an allelic difference from 138 to a maximum of 646 alleles (Figure 2). The closest related strains available from GenBank were *Ln. mesenteroides* strain 406 and 213MO, two strains isolated from Mongolian mare milk differing by 138 alleles, respectively, and 165 alleles from strain INF103 from producer E (Figure 2). CgMLST revealed 295 to 865 allelic differences between the Montenegrin isolates and the *Ln. mesenteroides* subspecies reference strains (Appendix A). CgMLST results were concordant with TYGS and ANI analysis (Appendix A, Appendix A, Appendix A, Figure 1).

### 3.4. Detection of Bacteriocin and Secondary Metabolite Genes

All isolates but one carried genes of the bacteriocin biosynthetic gene clusters, ten out of 16 carried a betalactone biosynthesis gene, and all carried a type III polyketide synthase gene (T3PKS) (Table 2). Eight out of our 16 *Leuconostoc* strains carried genes necessary for citrate metabolism (*cit*CDEFG operon) (Figure 3), 14 isolates had the acetoin reductase gene *butA* and all isolates carried genes necessary for the biosynthesis of branched aminoacids acetolactate synthase *alsS* and isoleucine synthesis operon *ilv*.

### 3.5. Safety Evaluation

Safety evaluation indicated that none of the 16 isolates carried pathogenic and virulence factors. Five isolates carried plasmids. The CARD database identified a C656T transition in the 23S rRNA gene of isolate INF36, conferring resistance to erythromycin and clindamycin (100% identity, 10% coverage) using the CARD database. ResFinder did not detect this mutation in the 23S rRNA gene of isolate INF36. Disk diffusion testing revealed susceptibility of strain INF36 against erythromycin (Table 3). Isolate INF117 carried a partial blaTEM with 99.04% identity and 60.51% coverage to blaTEM-141 (and others) using ResFinder. CARD analysis (using the loose hits criteria) confirmed the presence of a partial blaTEM in isolate INF117 (97.5% identity and 55.94% coverage to several blaTEM) (Table 3). A BLAST search of the contig containing the partial blaTEM was performed, resulting in 100% coverage and 99.65% identity match to *Ln. mesenteroides* ATCC8293. Disk diffusion testing revealed susceptibility of strain INF117 against ampicillin and penicillin (Table 3).

### 3.6. Orthofinder Analysis

Orthofinder analysis assigned 57,889,803 genes (97.9% of total) to 3080 orthologous groups. The number of orthologous groups identified for our isolates ranged from 1873 to 2111 (mean: 1979). There were 1280 orthogroups present in all *Leuconostoc* subspecies and 1091 of these orthologous groups consisted entirely of single-copy genes (Figure 3).

## 4. Discussion

Traditional food products are an acknowledged part of the identity, culture, and heritage of a country and, in addition, they may serve as a valuable resource for future food production [16,17,52]. To explore the diversity of *Leuconostoc* spp. Isolates, we characterized in this study 16 *Leuconostoc mesenteroides* (*Ln. mesenteroides*) isolates obtained from 13 traditional Montenegrin brine cheeses from nine different producers by whole genome sequencing (WGS). To the best of our knowledge, our study comprises the largest set of *Ln. mesenteroides* strains from traditional cheese characterized biochemically and by WGS.

The biochemical abilities of strains were in the range of expected technological significance. Growth at low/high salt and or low/high temperatures of single isolates could not be linked to genotypes.

Nevertheless, genome sequencing is considered as a suitable tool for the selection of particularly suitable strains used as adjunct cultures for the production of traditional cheese [15,21,22,26]. *Leuconostoc* species were “Generally Regarded As Safe” (GRAS) organisms by the US Food and Drug Administration (FDA) and the European Food Safety Authority (EFSA) for food production [53,54]. Reports about clinical infections caused by *Ln. mesenteroides* [55] in combination with the role of LAB as a reservoir for antimicrobial resistances have changed the uncritical classification of LAB as GRAS. Antimicrobial resistance and multi-resistance have been described for several *Ln. mesenteroides* isolates [36,56]. Safety evaluation of the Montenegrin brine cheese isolates revealed that they carried no pathogenic factors and no virulence genes. However, two isolates carried antimicrobial resistance determinants. One isolate carried a fragment of the blaTEM, which also exists in the genomes of several other *Ln. mesenteroides* strains including reference strains ATCC 8293. For the second strain, we found for the first time in a *Leuconostoc* spp. isolate a C656T point mutation in the 23S rRNA gene. This mutation was detected for the first time in *Clostridioides difficile* (*Clostridium difficile*) conferring high resistance to erythromycin and low resistance to clindamycin [57]. However, this point mutation was not found via ResFinder analysis. Phenotypic testing revealed that both strains were susceptible against the respective antibiotics, which can be explained by the fact that the detected antimicrobial resistance gene fragments are non-functional.

Analysis of genes essential for the biosynthesis of secondary metabolites revealed that all but one strain carried genes of the bacteriocin biosynthetic gene clusters, which is an important natural feature in cheese production through inhibiting the growth of several important pathogenic bacteria. All strains carried a type III polyketide synthase gene essential for the biosynthesis of important secondary metabolites [43] and all isolates carried genes essential for the biosynthesis of branched aminoacids that contribute to flavor. Interestingly, only 50% of our Montenegrin cheese isolates carried the cítrate operon, which is essential for citrate metabolism and a characteristic of dairy *Ln. mesenteroides* strains [20].

To investigate strain diversity, a previously published MLST scheme [25] and our novel ad hoc core cgMLST scheme were used. Our ad hoc cgMLST scheme comprised 960 core targets in comparison to a recently published cgMLST scheme, which comprised 999 core genes [19]. We used a so-called “hard core genome” definition and 19 instead of 17 complete genomes for core genome generation, which resulted in the observed lower number of 960 core genome targets. All tested isolates except four had more than 97% good core genome targets. Type Strain Server Analysis (TYGS) [38] identified most of our isolated strains as subspecies *Ln. mesenteroides mesenteroides,* confirming recent findings that subspecies *Ln. mesenteroides mesenteroides* is better adapted to cheese production than other subspecies and also than subspecies *Ln. mesenteroides cremoris,* which is mainly used in commercial starter cultures [3]. In addition to subspecies *Ln. mesenteroides mesenteroides*, TYGS analysis assigned the remaining six Montenegrin brine cheese isolates to subspecies *Ln. mesenteroides cremoris, *Ln. mesenteroides* jonggajibkimchii*, and *Ln. mesenteroides dextranicum.* Isolates identified as *Ln. mesenteroides mesenteroides* and *Ln. mesenteroides dextranicum* by TYGS clustered appropriately with the respective subspecies available from GenBank when using dDDH, ANI, or our cgMLST approach. Four isolates were not accurately assignable by TYGS, ANI, or cgMLST to a *Leuconostoc mesenteroides* subspecies. All these methods placed them between different Leuconostoc subspecies. Congruent with TYGS and ANI, cgMLST analysis placed two of the Montenegrin isolates between strains previously identified as *Ln. mesenteroides mesenteroides* [20] and *Ln. mesenteroides cremoris* [26]. Two other Montenegrin isolates were placed between *Ln. mesenteroides dextranicum* strains isolated from Italian cheese [26] and *Ln. mesenteroides mesenteroides* strains isolated from sheep milk in Slovakia [unpublished, https://www.ncbi.nlm.nih.gov/biosample/SAMN09398940] (accessed on 25 February 2020). From this Italian study [26], diverse *Ln. mesenteroides* subspecies were recently reassigned to *Ln. mesenteroides* lineage M4 and *Ln. mesenteroides* lineage M1 [20], which confirmed our core genome results, i.e., clustering of these strains on the same branches. In contrast, all other *Ln. mesenteroides cremoris* strains available from GenBank clustered together and were unrelated to the Italian and our isolates. This awkward subspecies assignment indicates that *Ln. mesenteroides* lineage M1 and M4 strains have some specific features from diverse subspecies [20]. The four Montenegrin isolates that were not assignable to a subspecies (and related to lineage M1 and M4 isolates according to Frantzen et al., 2017) [20] had, in comparison to the other Montenegrin cheese isolates, a lower percentage of good core genome targets (≤97%). This lower number of core genes is also characteristic for *Ln. mesenteroides cremoris* ATCC19254 and *Ln. mesenteroides jonggajibkimchii* DRC1506 due to the loss of genes of these two Leuconostoc subspecies as a consequence of adaptation to their ecological niches [19,20]. Our results support recent findings that a taxonomic revision of the genus *Leuconostoc* should be considered and that some of the isolates available in GenBank have been assigned to the wrong subspecies due to the use of phenotypic properties [20,58].

MLST and cgMLST analysis revealed that the 16 Montenegrin *Ln. mesenteroides* cheese isolates showed a high diversity, which contrasts with isolates obtained from commercial starter cultures, which show a low genetic diversity [20,25]. This high diversity underlines the importance of traditional food products as a valuable source for strains with unique and interesting features to be used in the dairy industry as novel starter cultures or for production of functional dairy foods [15,18,52]. The observed high diversity of our Montenegrin *Ln. mesenteroides* isolates and, in addition, their non-relatedness to genomes of strains available from GenBank reflects that no commercial starter cultures were used to produce the traditional Montenegrin brine cheeses included in this study. Although we investigated only a small number of isolates, the observed diversity corresponds with results from previous studies showing a high strain diversity of *Ln. mesenteroides* obtained from a variety of food sources [23,24,25]. Isolation of different strains from the same cheese product of the same producer and the diversity of strains obtained from the same cheeses also indicate a variable composition of the *Leuconostoc* population in traditional cheeses, which may also affect the sensory characteristics and quality of the final products. However, more extensive studies are necessary to prove these findings. The differences between isolates from the same producer as well as from the neighboring regions might be explained by the specific natural plant biodiversity, which is strongly supported by the fact that, in Montenegro, the feeding of animals is still to the highest extent based on natural grazing, with a relatively low percentage of concentrate feeding [52]. Thus, genome-based characterization of *Leuconostoc* spp. may present an innovative tool to prove the origin of strains and to ensure consistent manufacture of high quality and safe traditional food products [19,20]. In the future, with the ongoing progress in multi-omics technologies, the prediction of gene functions, metabolic pathways, and inter-microbe interactions—particularly within starter cultures—may allow a selection of strains based on specific marker genes more efficiently [28]. Currently, the accurate prediction of metabolic patterns and the comparison of metabolic capacities of organisms using genome-scale metabolic models is still a major challenge or even impossible due to the lack of complete genomes [8].

In conclusion, *Ln. mesenteroides* isolates from the Montenegrin traditional brine cheeses show a high genetic diversity, which can be explained by the high level of plant biodiversity in Montenegro and the fact that the feeding of animals is mainly based on natural grazing. WGS based strain-typing proves the Montenegrin origin and endogenous status of strains. With the ongoing progress in predictive microbiology, indigenous cheese isolates with specific properties will become selectable and become available for standardization of the production, to preserve designation of origin as well as to create added-value sensory characteristics.

## Figures and Tables

**Figure 1 microorganisms-09-01612-f001:**
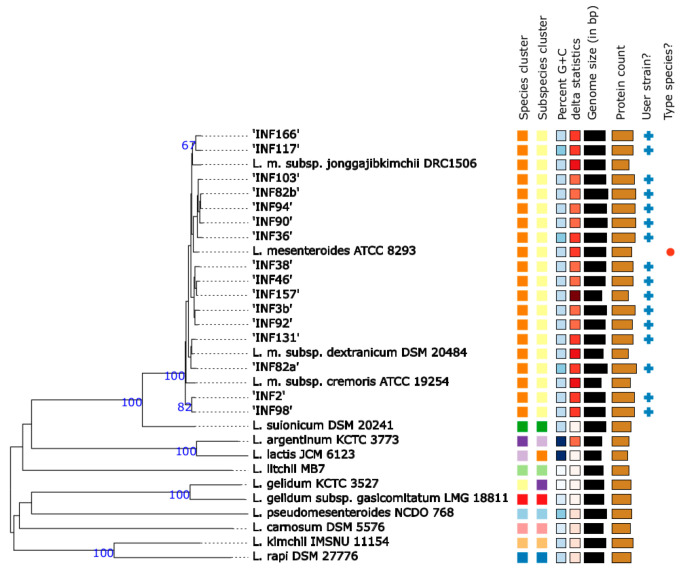
Type Strain Genome Server analysis (TYGS) of all 16 Montenegrin brine cheese isolates and *Leuconostoc* sp. type strains available from the TYGS database.

**Figure 2 microorganisms-09-01612-f002:**
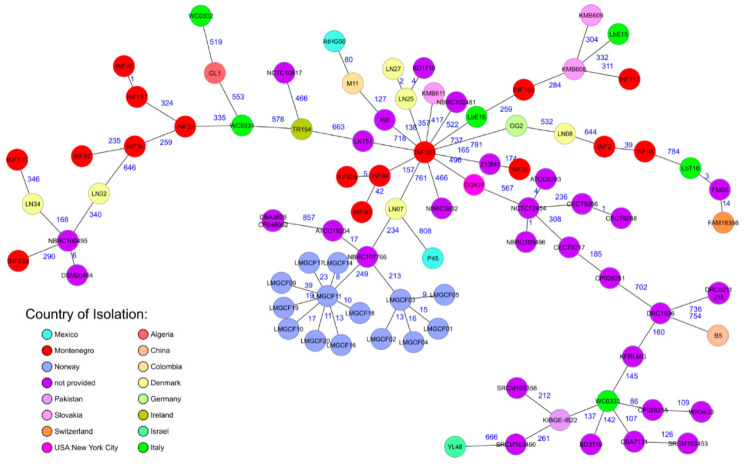
Minimum spanning tree (MST) based on cgMLST analysis of 88 *Ln. mesenteroides* isolates derived from GenBank and Montenegrin brine cheeses. Numbers on connection lines represent allelic differences between isolates. Isolates are colored according to their country of origin.

**Figure 3 microorganisms-09-01612-f003:**
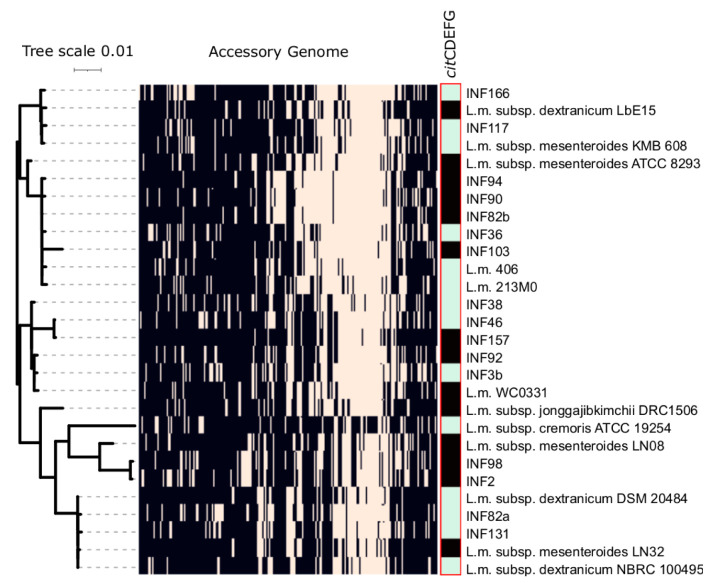
Schematic presentation of present (light colors) and absent (black) orthologous groups as determined by Orthofinder analysis and the *cit* operon (green) in Montenegrin brine cheese isolates.

**Table 1 microorganisms-09-01612-t001:** Data on origin, subspecies, growth, and biochemical parameters of the investigated *Ln. mesenteroides* strains. (nd) not done, (pg) poor growth, (cfu) colony forming units, (P) producer, (Cat) catalase (
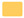
) pH decrease wo/w 1% lactose, (
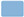
) pH increase wo/w 1% lactose, (

) *cit* operon positive strains, (
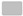
) no growth in minimum conditions, (
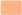
) growth in minimum and/or maximum conditions.

P	CheeseID	Strain ID	Subsp.-Id	Gram	Cat	EPS	CO_2_	pH Growth Medium after (h) wo/w 1% Lactose	Cfu/g Cheese	Growth with % NaCl	Growth °C
TYGS	(4 h)	(6 h)	(8 h)	(24 h)	(48 h)	2%	3%	4.5%	6.5%	4	10	15	30	45
**A**	2-1406	INF2	*Ln. m. cremoris*?	+	-	-	+	6.6/6.4	6.6/6.4	6.5/5.8	5.8/4.6	4.8/4.4	89	+	pg	-	-	-	+	+	+	-
3-1406	INF3b	*Ln. m. mesenteroides*	+	-	-	+	6.6/6.4	6.6/6.4	6.6/5.2	5.9/4.4	5.3/4.4	39	+	pg	-	-	+	+	+	+	- (+)
**B**	1-1404	INF36	*Ln. m. mesenteroides*	+	-	-	+	6.5/6.7	6.4/6.6	6.4/5.8	5.8/5.0	5.3/4.6	58	+	+	+	+	+	+	+	+	- (+)
2-1404	INF38	*Ln. m. mesenteroides*	+	-	-	+	6.5/6.5	6.4/6.3	6.4/6.1	5.9/5.2	5.8/4.9	61	+	+	+	-	-	+	+	+	-
**C**	1453	INF46	*Ln. m. mesenteroides*	+	-	-	+	6.4/6.6	6.4/6.6	6.0/6.1	5.7/5.4	5.4/4.7	41	+	+	+	+	+	+	+	+	+
**D**	1-3763	INF82a	*Ln. m. dextranicum*	+	-	-	+	nd	nd	nd	nd	nd	nd	nd	nd	nd	nd	nd	nd	nd	nd	nd
INF82b	*Ln. m. mesenteroides*	+	-	-	+	5.5/5.8	5.4/5.7	5.3/5.5	5.2/5.4	5.1/5.3	72	-	pg	+	pg	-	+	+	+	+
3-3763	INF90	*Ln m. mesenteroides*	+	-	-	+	5.8/6.0	5.9/5.8	5.8/5.4	5.7/5.3	5.7/5.1	53	+	+	+	+	-	+	+	+	+
INF92	*Ln. m. mesenteroides*	+	-	-	-	6.1/5.9	6.0/5.8	6.0/5.7	5.8/5.6	5.6/5.1	73	-	+	+	pg	-	+	+	+	+
INF94	*Ln. m. mesenteroides*	+	-	-	-	5.7/6.0	5.7/5.7	5.6/5.4	5.4/5.2	5.2/5.2	47	+	+	pg	pg	-	+	-	+	-
4-3763	INF98	*Ln. m. cremoris*?	+	-	-	+	5.4/6.0	5.4/5.9	5.4/5.8	5.2/5.8	5.2/5.5	63	+	+	+	pg	-	+	+	+	+
**E**	4-3758	INF103	*Ln. m. mesenteroides*	+	-	-	-	5.4/5.5	5.2/5.5	5.2/5.5	5.1/5.3	5.0/5.2	89	+	+	pg	pg	-	-	+	+	+
**F**	4-14246	INF117	*Ln. m. jonggajibkimchii*?	+	-	-	+	6.0/6.2	5.9/5.7	5.9/5.7	5.8/5.6	5.7/5.5	92	-	pg	+	+	-	+	+	+	+
**G**	**Ž**	INF 131	*Ln. m. dextranicum*	+	-	-	+	5.6/6.4	5.3/6.3	4.6/6.2	4.3/4.6	4.2/4.4	72	+	+	+	+	-	+	+	+	-
**H**	3-1387	INF157	*Ln. m. mesenteroides*	+	-	-	+	6.4/6.6	6.3/6.6	6.2/5.8	5.8/5.5	5.4/4.9	41	+	+	+	+	-	+	+	+	+
**I**	4-1383	INF166	*Ln. m. jonggajibkimchii?*	+	-	-	+	6.7/6.4	6.5/6.4	5.8/5.9	5.1/4.6	4.8/4.4	42	+	+	+	-	-	+	+	+	-

**Table 2 microorganisms-09-01612-t002:** Genes of the bacteriocin gene cluster and biosynthesis of secondary metabolites identified by BAGEL4 and AntiSMASH.

Isolate-ID	Bacteriocin	Betalactone	T3PKS	Other
INF2	IIc	+	+	Enterocin_x_chain_beta
INF3b	IIc, MesentericinY105	-	+	Enterocin_x_chain_beta
INF36	IIc	+	+	Enterocin_x_chain_beta
INF38	IIc	+	+	Enterocin_x_chain_beta
INF46	IIc	+	+	Enterocin_x_chain_beta
INF82a	IIc	+	+	Enterocin_x_chain_beta
INF90	IIc	+ LomaiviticinA//C-E	+	Enterocin_x_chain_beta
INF82b	IIc	LomaiviticinA//C-E	+	Enterocin_x_chain_beta
INF94	IIc	LomaiviticinA//C-E	+	Enterocin_x_chain_beta
INF92	IIc	-	+	Enterocin_x_chain_beta
INF117	IIc	-	+	-
INF103	IIc	-	+	-
INF98	IIc	-	+	Enterocin_x_chain_beta
INF131	MesentericinB105	+	+	Enterocin_x_chain_beta
INF157	IIc	-	+	Enterocin_x_chain_beta
INF166	-	+	+	-

**Table 3 microorganisms-09-01612-t003:** AMR targets, virulence genes, plasmids, pathogenicity, (S) susceptible, (Ery) erythromycin, (Amp) ampicillin, (Pen) penicillin, (Inhz) inhibition zone, (-) negative, and (nd) not done.

Strain ID	AMR			Plasmid Finder	Virulence Finder	Pathogen Finder
	CARD	ResFinder	Susceptibility Testing			
INF2	-	-	nd	Rep3	-	-
INF3b	-	-	nd	Rep3	-	-
INF36	Clostridioides difficile 23S rRNA with C656T mutation conferring resistance to erythromycin and clindamycin, 98.97% identity, 10.00% coverage	-	S Ery (15 µg) Inhz 26 mm	-	-	-
INF38	-	-	nd	-	-	-
INF46	-	-	nd	-	-	-
INF82a	-	-	nd	-	-	-
INF82b	-	-	nd	-	-	-
INF90	-	-	nd	-	-	-
INF92	-	-	nd	Rep3	-	-
INF94	-	-	nd	-	-	-
INF98	-	-	nd	Rep3	-	-
INF103	-	-	nd	-	-	-
INF117	97.5% identity to blaTEM 33, 55.94% coverage (using loose criteria)	beta lactam resistance 99.04% identity to blaTEM 141, 60.51% coverage	S Amp (10 µg) Inhz 19.5 mm Pen (1U) Inhz 21 mm	-	-	-
INF131	-	-	nd	-	-	-
INF157	-	-	nd	-	-	-
INF166	-	-	nd	RepA_N	-	-

## Data Availability

This *Leuconostoc mesenteroides* whole genome shotgun (WGS) project has been deposited in DDBJ/ENA/GenBank under the BioProject No. PRJNA706746. This Whole Genome Shotgun project has been deposited at DDBJ/ENA/GenBank under the accession JAGFYI000000000—JAGFYL000000000, JAGSYL000000000, JAFREA000000000—JAFREC000000000, JAFRDS000000000—JAFRDZ000000000. The version described in this paper is version JAGFYI010000000—JAGFYL010000000, JAGSYL010000000, JAFREA010000000—JAFREC010000000, JAFRDS010000000—JAFRDZ010000000.

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
