# Peer review of "Genetic Diversity of Leuconostoc mesenteroides Isolates from Traditional Montenegrin Brine Cheese"

_microorganisms, 2021, doi:10.3390/microorganisms9081612_

Round 1

Reviewer 1 Report

This manuscript (MS) deals with genetic diversity of sixteen Leuconostoc mesenteroides strains isolated from various traditional Montenegrin brine cheeses. The biochemical and genetic characterizations of the strains were compared through various experiments and computational analysis methods. Although there are many previous studies on the genetic diversity of lactic acid bacteria in fermented foods, the data of this study seems valuable because studies based on whole genome sequencing of multiple lactic acid bacteria are very rare. Nevertheless, compared to the experimental results, the text content is very poor. Moreover, there are many spacing errors and typos. Therefore, this MS should be supplemented and refined for possible publication.

Major comments

  • It is strongly recommended to divide the results part into some subparts.
  • L. 231-243 and Table 1: The description of the results on biochemical properties experiments is too concise and therefore, it is difficult to understand. For example, the method and result for the viable cell count experiment were missed, so I do not know meaning of the experiment. Table 1 should be explained in much more detail.
  • Figure 1 does not appear to be significant because there is no correlation between genetic features and regions which the cheese is isolated. It is recommended to move to supplementary part.
  • Table 1: INF82a was not grown in all conditions tested. How did you grow that strain and extract the genome?
  • Figure 3: The resolution should be improved.
  • Line 336-338 and Table 2: There are many kinds of bacteriocin. What is the bacteriocin? In addition, there is no explanation for the T3PKS, citCDEFG operon, butA, alsS and ilv. It should be explained what the function of each gene is and why the genes were selected and compared.
  • L. 340-353: Antibiotic resistance genes were found in INF36 and INF117. Have you checked whether the two strains are actually antibiotic resistant?

Minor comments

  • L. 137: ‘560’ should be indicated as a subscript.
  • L. 139: Subheading number was missed.
  • L. 234-235: 24h – 24 h...
  • L. 241: INF3b(pro – INF3b (pro
  • L. 242: INF 36 – INF36
  • Table 1: Does 6,6 mean pH 6.6? If so, all commas should be replaced with periods.
  • Table 1: vcc – CFU
  • L. 261: 2.0Mb, 2.3Mb – 2.0 Mb…

Reviewer 2 Report

Thanks for your research. It was an easy work to read. The era of omics has provided valuable tools that generate a lot of information during a scientific investigation. In the case of genomic analyzes, the information obtained is greatly enriched by evaluating those activities that are active and which only remain contained in the genes. What can we do with this information obtained?

L-135: Why in the rest of the determinations did the strains grow at 30 oC and in the salt tolerance test they did so at 37 oC?

153: EPS, the determination in the EPS production is very subjective, there are different quantitative methodologies, for example, DOI: 10.13057/biodiv/d210623,

DOI: 10.1039/c7ra03925e

Figure 1 does not provide relevant information to the results.

It was very difficult to differentiate the colors used in figure 3.

It is very interesting to know that the identified strains contain the genes to produce bacteriocins (Table 2). The information obtained from the genomic analysis can be complemented with an analysis of the antagonistic activity in vitro.

L-403_405: But it does not mean that they exhibit antimicrobial activity. Therefore, this analysis must be carried out, which can provide very interesting information.

Reviewer 3 Report

Overview:

The manuscript describes the characterisation of 16 strains of Lc. mesenteroides isolated from Montenegrin brine cheeses manufactured in different regions of the country.  The analysis provided is extensive (using whole genome sequencing, analysis of the core genomes and comparisons with Leuconostoc species isolated from other fermented foods) and thorough.  The data in the text and supplementary materials is clearly presented on the whole (see minor comments to assist in clarifying the shading in tables/diagrams, annotated on the reviewed pdf).  The conclusions are supported by the data.  However, the survey of genetic diversity is based on only 16 strains so drawing a conclusion that there are differences between plants (seasonality etc) is not totally valid, as other published surveys of different LAB have been based on many more isolates – often in the hundreds (albeit with lesser genomic analysis).  The conclusion is in line with prior publications of LAB surveys, which have shown seasonal, location and manufacturing plant differences in the major LAB species detected.  The value of the research is in increasing the genomes available for future comparative work and will be of interest to other researchers in this field.

Editorially:  the manuscript is clearly presented and minor editorial matters will need to be addressed in the final version.  See comments herewith and on the annotated pdf.

Abstract:

The genes mentioned as producing aromatic compounds are ilv and alsS, which are involved in branch-chain amino acid synthesis and acetolactate metabolism respectively.  Please review and correct.

Introduction:

Line 35: Please use the abbreviation for Leuconostoc (Lc.) for all species named not just defining this for the last species in the list.

Line 45:  Leuconostic spp are (spp is plural)

Line 75:  journal format for citing references? I presume that all of the format matters will be reviewed in revision/editing, including citation of references and format of section headings

Methods:

Lines 100 and 109:  How were the bacteria isolated? selective media used?  Details are required or a reference to prior publication of the isolation of these strains is needed.  There is a variety of selective media for different types of NSLAB so it is important to understand how the bacteria were selectively, or non-selectively, isolated – for the record.

Lines 111-112: A reference is need to demonstrate how this method is applied to this genus

Line 185: Do you mean 'a multilocus sequence typing scheme developed in this work, based on the core genome of Lc. mesenteroides'?  Later in the manuscript the authors refer to a similar extensive MLST scheme with over 960 genes considered but the ‘ad hoc’ scheme used differed.  This is why it is appropriate to indicate clearly that the scheme used was developed by the authors and based on prior similar/published research.  Please review and clarify, if this is the case, or simply say that this is described in the following sentences – which it is.

See marked-up pdf for other queries regarding format.

Results:

Table 1:  make sure that all of the parameters/colours are defined e.g. ‘P’, what are the colours in the right hand side of the body of the table?

Line 257: The authors need to state somewhere what ANI cut-off point was used to determine species identity - 95% is generally accepted and the supplementary data shows >97% ANI similarity, so all 16 strains are, as contended, Lc. mesenteroides.

Table 3:  Make sure the colours are defined.  This is clear but an explanation is needed.

Discussion:

The survey conducted was only on 16 isolates so this is not very extensive study, despite it thoroughness.  Consequently, this conclusion (lines 457-459) is a little overstated here - diversity seen in 16 strains supports the contention that the strains vary between plant and region but more extensive published surveys of other cheeses/species have involved hundreds of strains.  The methodology described here (MLST, whole genome sequencing) is solid and the analysis is probably more extensive than prior publications.

Author Response

Please see the attachement!

Round 2

Reviewer 1 Report

The authors made appropriate revisions according to this reviewer's suggestions.

Author Response

Thank you, we have corrected/ accepted all the minor comments from the editor!

KR